# Preclinical Models of Neuroblastoma—Current Status and Perspectives

**DOI:** 10.3390/cancers15133314

**Published:** 2023-06-23

**Authors:** Ewa Krawczyk, Joanna Kitlińska

**Affiliations:** 1Department of Pathology, Center for Cell Reprogramming, Georgetown University Medical Center, Washington, DC 20057, USA; 2Department of Biochemistry and Molecular & Cellular Biology, Georgetown University Medical Center, Washington, DC 20057, USA

**Keywords:** neuroblastoma, in vitro cancer models, animal cancer models, preclinical disease models

## Abstract

**Simple Summary:**

Neuroblastoma is a pediatric tumor originating from the precursors of sympathetic nerves. The disease is known for its high heterogeneity. Hence, developing adequate preclinical models reflecting the complex biology of neuroblastoma is particularly challenging. This paper describes the current status of the available neuroblastoma models with their strengths and limitations, and demonstrates the future perspectives for preclinical neuroblastoma research.

**Abstract:**

Preclinical in vitro and in vivo models remain indispensable tools in cancer research. These classic models, including two- and three-dimensional cell culture techniques and animal models, are crucial for basic and translational studies. However, each model has its own limitations and typically does not fully recapitulate the course of the human disease. Therefore, there is an urgent need for the development of novel, advanced systems that can allow for efficient evaluation of the mechanisms underlying cancer development and progression, more accurately reflect the disease pathophysiology and complexity, and effectively inform therapeutic decisions for patients. Preclinical models are especially important for rare cancers, such as neuroblastoma, where the availability of patient-derived specimens that could be used for potential therapy evaluation and screening is limited. Neuroblastoma modeling is further complicated by the disease heterogeneity. In this review, we present the current status of preclinical models for neuroblastoma research, discuss their development and characteristics emphasizing strengths and limitations, and describe the necessity of the development of novel, more advanced and clinically relevant approaches.

## 1. Introduction

Preclinical disease models play crucial roles in all fields of biomedical research, including drug discovery and development, the implementation of drug screening, diagnostic tests, prophylactic and therapeutic vaccines, evaluation of disease mechanisms, and discovering biochemical pathways. Despite their limitations, these models have been indispensable in cancer research conducted before clinical trials. However, none of the preclinical cancer models are ideal. The extent of the clinical relevance of the existing models has been a long-standing problem in cancer research [1,2,3,4,5].

Classic models, such as in vitro two-dimensional cell cultures and laboratory animals, have been extensively used in cancer research for decades [6]. Various established cell lines, including HeLa, are most commonly used, since they are publicly available, inexpensive, easy to handle, and typically provide highly replicable results. Unfortunately, the erroneous use of cells in laboratories around the world is surprisingly frequent—the cross-contamination rate can be as high as 25%, and numerous cell lines can be misidentified and mislabeled [7,8,9,10], which jeopardizes the research quality. However, even if used under their true identity, after many years of passaging the established cancer cell lines often no longer resemble the original cancers. They undergo changes that make them different from the source tumor: genetically, morphologically, metabolically, and physiologically. One of the reasons for this phenotypic drift is culture on plastic, in 2D setting, lacking the structure and microenvironment of the tissue of origin [5,11]. Moreover, while tumor cell lines can be grown in laboratories for a long time, the same cannot be conducted with corresponding normal cells, derived—as a necessary control—from patients’ normal tissues [12].

Mouse models of human cancer are not without problems as well. Ideally, they should recapitulate the events occurring in a patient and mimic the pathology, genetics, and therapeutic response of human disease. Neither classic mouse models, such as transgenic mice and conventional knockouts, nor more advanced ones (e.g., conditional knockouts and mice with regulated expression of oncogenes) fulfill all the requirements for a perfect model. The mouse organism resembles its human counterpart in many aspects, but there are also important differences (e.g., the control of telomeres and telomerase) that affect and limit potential research and the relevance of the obtained results to clinical practice [13,14,15,16].

Because of the limitations of classic preclinical models, new advanced models are being developed and extensively utilized. Various in vitro techniques such as three-dimensional cell cultures including gel embedding, scaffold culture, hanging drop culture, microfluidic devices, 3D bioprinting [7,17,18,19,20], induced pluripotent stem cells [21,22], and patient-derived 2D cell culture methods [23,24,25,26,27]. Recent years also brought significant advances in animal models, including humanized mouse models and patient-derived xenografts (PDXs) [28,29]. Importantly, none of these techniques fully recapitulate the process of carcinogenesis and cancer progression. Hence, they can be optimized and combined to complement each other, and thereby provide reliable tissue and organ models [30,31,32,33,34]. Currently, morphologically and genetically accurate complex in vitro models (CIVMs) are being developed to enable studies on particular cancer types [35].

Neuroblastoma is a rare solid cancer of the sympathetic nervous system, derived from neural crest cells. It is the most common childhood malignancy; the median age of diagnosis is 18 months and most of the cases occur in children below 10 years of age. Neuroblastoma is a very heterogenous disease, with a diverse clinical presentation and molecular complexity [36,37,38,39,40,41]. Despite an advancement in therapy in recent years, the prognosis for high-risk patients remains poor [42]. Most high-risk neuroblastomas initially respond to the therapy, but eventually relapse; the fatality rate of the disease is 50%. These unfavorable outcomes are commonly associated with amplification of the *MYCN* protooncogene (encoding N-MYC transcription factor), mutations of the anaplastic lymphoma kinase (*ALK*), and segmental chromosomal alterations (e.g., loss of chromosome arm 1p and 11q, gain of 17q) [37,43,44]. Moreover, patients with neuroblastoma may experience various neurological symptoms, metabolic syndrome, growth and puberty impairment, secondary neoplasms, and other syndromes, depending on the location of the primary tumor and metastases [40,45]. Hallmarks of neuroblastoma—heterogenous and diverse clinical manifestations, poor prognosis of advanced disease and high metastatic frequency—require new strategies in drug discovery and administration [46,47]. Advanced preclinical models are indispensable in that approach, as well as in attempts for a better understanding of the mechanisms and biology of the disease [36,48]. Our contemporary review summarizes the current status as well as recent developments in preclinical models used in various fields of neuroblastoma research, both in vitro and in vivo. We emphasize the importance of the clinically relevant models, especially for a rare disease such as neuroblastoma, discuss the strengths and limitations of each model, and describe the future advances in this field.

## 2. In Vitro Models of Neuroblastoma

### 2.1. Conventional 2D Cell Cultures

Currently, neuroblastoma research relies largely on cell lines cultured in vitro in two-dimensional settings. Numerous established neuroblastoma cell lines of human or rodent origin are available commercially (Table 1) [49,50]. Despite the disadvantages, the cells maintained using this technique are easy to use, and the method is highly productive and inexpensive. They preserve the distinct chromosomal aberrations that are characteristic for neuroblastoma and contribute to the disease prognosis and patient outcome [51,52]. Recent advances in CRISPR/Cas9 technology allow for further genetic modification aiming at recapitulating such genomic changes and thereby testing their functions [53]. Moreover, neuroblastoma cell lines recapitulate heterogeneity in differentiation stages observed in human tumors, ranging from undifferentiated mesenchymal to committed adrenergic cells [54,55]. For these reasons, the established cell lines are broadly used in various fields of neuroblastoma research, including basic tumor biology and interactions with the tumor microenvironment [56,57,58], as well as the development of novel treatment strategies, treatment screening, and monitoring [47,59,60,61]. Moreover, the neuroblastoma cell lines have been used in a wide spectrum of studies on genetic, biochemical, functional, and structural neuroblastoma characteristics, for example on the interaction of neuroblastoma cells with Schwann cells, the synthesis of neurotransmitters, differentiation and transdifferentiation processes, chromosomal structure, the role of *MYCN* expression, the importance of neuropeptide Y and hypoxia, and many others [57,62,63,64,65,66,67,68,69,70,71,72,73,74,75,76,77,78,79,80,81,82,83]. A recent CRISPR/Cas9 screen performed in a panel of neuroblastoma cell lines has led to the identification of EZH2 as a molecular driver of the *MYCN*-amplified neuroblastomas, paving the way for similar studies in the future [84], importantly, in 2D cell lines, potential chemotherapy [85,86,87,88,89], radiation [90], and immunotherapy [91,92,93,94] screening. Furthermore, they are utilized to determine the role of genetic and chromosomal alterations in tumor growth, development, and the possible prognosis of the disease [95,96,97,98].

Established neuroblastoma cell lines have also been used for validating the mechanistic models used for the prediction of neuroblastoma progression based on the analysis of molecular networks [99]. Due to their neuronal features, neuroblastoma cell lines are also used as models in neuroscience, e.g., in studies on Alzheimer’s disease [100], Parkinson’s disease [101], and in virology research [102,103].

Despite the rarity of the disease, the field of neuroblastoma research is equipped in a vast number of established cell lines. This includes unique sets of cell lines developed from the same patient before diagnosis and post-treatment, rarely available for other cancer types [104]. This includes SK-N-BE(1) and SK-N-BE(2), SMS-KAN and SMS-KANR, SMS-KCN and SMS-KCNR, as well as CHLA15 and CHLA20 cell lines. Moreover, recent years brought significant progress in the development of new neuroblastoma research tools. Thanks to collaborative efforts led by the Children’s Oncology group, a wide range of cell lines developed from different biological materials (tumors, bone marrow, or blood), at different stages of the disease (at diagnosis, post-treatment, and post-mortem) is now available through the Childhood Cancer Repository [105]. This collection of well-characterized neuroblastoma cell lines reflecting neuroblastoma progression and representing its most aggressive, therapy-resistant, and metastatic features provides an excellent experimental tool for designing and testing novel therapeutic strategies.

Although most commonly used in cancer research, the conventional 2D cell culture is associated with significant disadvantages. Even though freshly obtained neuroblastoma tumor initiating cells retain certain tumor characteristics in cell culture [106], it has been known for years now that in vitro cultivation causes an accumulation of numerous alterations in cells [107]. The cells may undergo rapid epigenetic and transcriptional changes [108], differentiate [109], lose the expression of certain genes [110], change the biophysical properties of the cell membrane [111], lose sensitivity to oxidative stress [112], and undergo many other alterations. Conventional 2D cell lines lack polarity and their contact with tumor environment is affected [4], they are chromosomally unstable [3], and they do not preserve the heterogeneity and complexity of the tissue of origin. The heterogeneity of the tumor is currently described not only as a difference between the cancer cells in morphology, transcriptional profiles, and metabolism, but also includes the differences between the tumor microenvironment, which plays an important role in tumor resistance to therapy, the higher level of metastasis, and aggressiveness [113,114,115]. With two-dimensional cell lines, we are not able to properly reproduce these conditions in vitro [2,5,11,26,116].

In general, neuroblastoma cell lines, similar to other conventional cancer cell lines, do not properly recapitulate the original tumor properties and do not fully reflect the complexity and heterogeneity of the malignancy [117]. Thus, they may not be the ideal model for preclinical studies as the results obtained with them may not be transferable to patients. Therefore, various approaches to solve the problem with conventional 2D cell cultures in preclinical studies have been attempted. These include various three-dimensional cultures, co-cultures with additional cells, and complex models with microfluidic perfusion systems [118,119].

**Table 1 cancers-15-03314-t001:** Examples of widely used neuroblastoma cell lines commercially available from American Type Culture Collection (ATCC).

Name	Origin	Stage (INSS *)	Treatment	MYCN Status	ALK Status	P53 Status	Differentiation Status	References
IMR-32	Human		none	amplified	wt	wt	adrenergic	[49,54,85,120,121]
SK-N-SH	Human	4	CT/RT	non-amplified	mut	wt	adrenergic	[54,85,105,122,123,124]
SK-SY5Y	Human; thrice cloned (SK-N-SH -> SH-SY -> SH-SY5 -> SH-SY5Y) subline of SK-N-SH	4	CT/RT	non-amplified	mut	wt	adrenergic	[50,54,85,124,125,126]
SK-N-BE(2)	Human	4	CT/RT	amplified	wt	wt		[49,85,125,127]
BE(2)-C	Human; clone of SK-N-BE(2)	4		amplified			adrenergic	[54,125,128]
BE(2)-M17	Human; clone of SK-N-BE(2)	4						[125]
Neuro-2a	Mouse	-						[79]
SK-N-FI	Human	4	CT	non-amplified	wt	mut	adrenergic	[54,77,85,105]
SK-N-DZ	Human	4		amplified	wt	wt		[78,85,123]
B35	Rat	-						[66]
SK-N-AS	Human	4		non-amplified	wt	wt	mesenchymal	[50,54,77,123,124]
N1E-115	Mouse	-						[65]
NBFL	Human	4						[64]
CHP-212	Human			amplified				[49,128]
NB41A3	Mouse	-						[62]

* International Neuroblastoma Staging System [129]; CT: chemotherapy; RT: radiotherapy; wt: wild type; mut: mutant.

### 2.2. Conditionally Reprogrammed Cells

Conditional reprogramming as the novel method of cell culture has been described in 2012 [130]. This method allows growing the epithelial cells, both cancerous and normal, for an indefinite time in a 2D in vitro setting without the transduction with any exogenous genes. The proliferative and adult stem-like state of the conditionally reprogrammed cells is maintained by a co-culture with irradiated mouse fibroblasts as the feeder cells, and by the presence of the Rho-associated kinase (ROCK) inhibitor, Y-27632 [25]. These cells are karyotype-stable, the induction of conditional reprogramming is rapid and reversible [131]. Conditional reprogramming is the epitome of personalized medicine; the cells can be directly obtained from a patient, cultured, and used for potential therapeutic approaches [23]. They can be also used as models for certain cancer types, including rare diseases, e.g., neuroendocrine cervical carcinoma [24,27,132], as well as biobanked for future basic and preclinical applications, for example genetic and chemosensitivity testing [133,134,135,136]. Conditional reprogramming is an inexpensive, easy to use, and robust method [25]. It can be used for growing normal cells [26,137], expanding cells generated from PDXs [30], as well as growing animal cells [138]. It has been shown that conditionally reprogrammed cells obtained from mammary tumors retain the genetic characteristics of the source malignancy [139] and maintain the expression of the estrogen receptor ERα; a well-known challenge in breast cancer research [140]. Undoubtedly, the main disadvantage of conditional reprogramming is that being a two-dimensional setting, it does not preserve the structural organization of a tumor.

We have shown that conditional reprogramming can be used to generate, culture, and biobank neuroblastoma cell lines [141]. Murine neuroblastoma conditionally reprogrammed cells collected from tumors arising in TH-MYCN mice retain the characteristic heterogenic neuroblastoma phenotype including a mesenchymal and neuronal component [54] (Figure 1A), and are useful as a neuroblastoma cell model in basic research [67]. These cells may also be grown in 3D cell culture settings (Figure 1B–D), thus serving as an example of the combination of various preclinical models. We propose that conditional reprogramming—when applied to human neuroblastoma samples—may serve as a novel reliable model for basic and translational research, including personalized drug testing.

### 2.3. 3D Cell Cultures

Cancer is a complex, multidimensional disease, so it cannot be adequately represented in a 2D cell culture setting. Classic 2D cultures lack the microenvironment of the tumor and the complex structure of the tissue as whole. On the other hand, when animals are used as preclinical models, their genetic and pathophysiological characteristics are different from the human body in many ways. Moreover, animal research is associated with ethical issues, as the number of animals used for biomedical experimentation should be reduced as much as possible [142]. Therefore, in an attempt to create better preclinical cancer models, three-dimensional cell cultures are becoming increasingly utilized. This includes spheroids and organoids in various matrices, as well as more complex settings, such as microfluidic systems or material engineering for the scaffolds [33,143,144]. In cancer research, 3D in vitro models are particularly useful in representing certain cancer-specific hallmarks, including the replication potential, tissue invasion, and response to growth signals [116,145,146].

Spheroids are maintained in a scaffold-free and gel-free setting: they are grown in the hanging drop culture or on low-attachment plastic surfaces. The cells aggregate spontaneously, and the aggregates are composed of different subpopulations of cells, thus expressing the structural heterogeneity important for studies aimed for discovering anticancer therapy or the identification of pathways involved in tumor biology and cancer biomarkers [18,147,148,149]. The 3D multicellular cultures in vitro have many advantages over traditional cells maintained in the 2D setting, as they recapitulate the cellular complexity and heterogeneity of a tumor and the well-defined structural organization of the colony [150]. Spheroid models are also utilized in various aspects of neuroblastoma research, including the development and monitoring of combination chemotherapy [59,151,152,153], stem cell research [154], radiobiological studies [155], and the exploration of tumor growth determinants and their complexity [156]. Special microgravity-assay bioreactors, promoting spontaneous cellular interactions and aggregations and thus permitting formations of spherical 3D cell cultures, have also been useful for propagating neuroblastoma cells [157].

The local microenvironment, specifically its main component, the extracellular matrix (ECM), plays a crucial role in cancer development. It controls almost every aspect of cellular behavior, being involved in the regulation of processes such as cancer invasion, tumor-associated inflammation, cell polarity, and cancer stem cell niches [158]. Therefore, gel-embedded 3D cultures are thought to more accurately mimic in vivo tumor conditions, and they are arguably considered a better in vitro tumor model than growth in suspension. Matrigel, collagen, alginate, and other emerging gel techniques, such as engineered scaffolding structures [159,160,161], provide adequate structural support for cells, which allows for more precise modeling interactions between tumor cells and ECM. This, in turn, results in clinically relevant outcomes, such as testing the response to anticancer drugs [20,145]. These organoid models have been used for breast cancer cells [17], urological cancers [162], lung cancer [163], brain tissue [164], and many others [20,145,165,166]. Their strengths include the applicability to numerous research fields, from basic science to modelling diseases, including development and infectious diseases, drug discovery and screening, as well as personalized medicine [33,167]. They are versatile, expandable, genetically stable, and can be adapted for gene-modification techniques. However, they are costly and time-consuming.

The 3D gel-embedded cell cultures, utilizing for example the collagen-based scaffolds supplemented with nanohydroxyapatite or glycosaminoglycans [168] or composite hydrogels based on gelatin or alginate [118,169,170], are tested or used for neuroblastoma as well [171], to study the heterogeneity of the tumor [98], immunotyping [172], and chemosensitivity [173,174]. They are expected to improve the correlation between the results obtained in vitro and the outcomes in patients.

### 2.4. Complex In Vitro Cultures

Recently, a novel 3D bioprinting technique that recapitulates the tumor microenvironment and the spatial distribution of the cells as well as allows for the addition of other types of cells, has been applied to neuroblastoma cells [118,119,175]. To evaluate the role of vascularization and angiogenesis in neuroblastomas, an advanced model mimicking neuroblastoma vasculature was established [176,177,178,179]. These models may allow for advanced studies on neuroblastoma structure, progression, angiogenesis, and metastasis. In the near future, they may prove to be useful, providing a precision medicine platform for the assessment of potential therapeutic options.

### 2.5. Models of Neuroblastoma Initiation

The culture of neuroblastoma cells derived from human and mouse tumors serves as a useful tool to test the disease phenotypes and response to treatment; however, it does not recapitulate the early events leading to the disease initiation during sympathoadrenal development. To fill this gap, researchers have utilized in vitro techniques mimicking normal sympathetic differentiation, in the presence or absence of genetic aberrations observed in human neuroblastomas. These approaches include the differentiation of human and mouse embryonic stem cells, as well as human induced pluripotent cells into the neural crest, sympathoadrenal progenitors, or sympathetic nerves [180,181,182,183]. Others propose the use of human neural progenitor cells obtained by the direct reprogramming of the somatic cells, as well as the isolation of sympathetic neuroblasts from chick embryos or sympathoadrenal progenitor cells from postnatal murine adrenal glands [90,180,184]. Thus far, the number of available models of neuroblastoma initiation is limited and the methodology is still being developed. However, as the interest in the dysregulation of developmental processes leading to malignant transformation is growing, the above techniques become more commonly used in research on neuroblastomas and other neuronal tumors [182].

## 3. Animal Models of Neuroblastoma

Rodent cancer models, especially murine models, have been utilized in neuroblastoma research for decades to explore the genetics and mechanisms of the disease, as well as treatment and diagnostic options. The ability to recapitulate multiple processes involved in cancer progression and interactions between tumor cells and their environment are advantages of animal models. However, such experiments are typically costly, time-consuming, and not suitable for high-throughput studies. Commonly used murine models of neuroblastoma are classified into syngeneic, transgenic, xenograft, and humanized animal models [117]. The strengths and limitations of the most commonly used in vivo neuroblastoma models are summarized in Table 2.

### 3.1. Genetically Engineered Mouse Models

The best characterized alteration in human neuroblastoma and the most important predictor of its poor prognosis is the amplification of a *Myc*-related gene, *MYCN*. Moreover, MYCN directly influences the genetic and epigenetic changes observed in neuroblastoma [185]. Because of that, *MYCN* has been extensively used for modelling neuroblastoma in mice. In 1997, Weiss and colleagues created transgenic mice that overexpress *N*-*myc* in neural crest lineage cells under the control of rat tyrosine hydroxylase gene (TH) promoter [186]. Transgenic TH-MYCN mice demonstrate a high rate of neuroblastoma tumors that closely resemble human disease, based on their histology, pathology, molecular biology, and location in the paraspinal sympathetic ganglia [187,188]. The tumors also exhibit the heterogeneity observed in human neuroblastomas, with both adrenergic and mesenchymal populations present within the tumor tissue [141]. However, no tumors are observed in the adrenal glands of TH-MYCN mice, which is a common location of human neuroblastoma. Moreover, despite a high MYCN expression, which in humans correlates with metastatic disease, no overt metastases are seen in the TH-MYCN model. While disseminated tumor cells are observed in the lungs, no metastases in typical neuroblastoma locations, i.e., bones, bone marrow, or liver, are observed [189]. Lastly, the experiments on these mice are complicated by a long time for tumor development and the lack of the full penetrance, with the reported tumor frequency in hemizygous mice between 30% and 50% [186]. Nevertheless, the TH-MYCN murine model—as well as derived modified models (e.g., TH-*MYCN*/*Mdm2*^+/−^, TH-*MYCN*/TH-Cre/*Casp8*^flox/flox^, LSL-*MYCN*; Dbh-iCre, TH-*MYCN*/*Trp*53^KI/KI^), which can overcome some issues associated with the original TH-MYCN mice—are extensively utilized in various fields of neuroblastoma research. These include testing the potential chemotherapeutics [187,190,191,192,193,194,195,196], the identification of biomarkers [197,198], and a study on various aspects and mechanisms of tumorigenesis, and metastasis, as well as the development and progression of neuroblastomas [43,195,199,200,201,202,203,204,205].

In addition to MYCN amplification, aggressive neuroblastoma is also associated with activating mutations of the ALK oncogene and the overexpression of the epigenetic regulator involved in development, LIN28B. While some attempts of creating a mouse model expressing either the most common ALK mutation in neuroblastoma, ALK F1174L [206], or overexpressing LIN28B [207] in sympathetic lineage resulted in the formation of neuroblastoma tumors, other studies indicated that these genetic lesions alone are not capable of triggering a neuroblastoma. Instead, in these models, the expression of ALK mutants [208,209] or LIN28B [210] resulted in increased tumorigenicity, when combined with MYCN overexpression, as seen in TH-*MYCN*/ALK^F1174^ mice. Nevertheless, the genetic changes detected in the above models correlate well with the genetic landscape of human neuroblastomas, making them an adequate model to investigate the perturbations of developmental processes leading to neuroblastoma initiation and the therapy response of these tumors [196,208,211,212]. Unfortunately, similarly to the original TH-MYCN mice, none of these models accurately recapitulate the metastatic processes observed in neuroblastoma patients, as the tumors developing in the transgenic mice spread rarely and do not show tropism to the metastatic niches most common in neuroblastoma patients. Recently, a new model with c-MYC overexpression has been reported (*Dbh*-*iCre*/*CAG*-*C*-*MYC* mice); however, the details on the tumor frequency and disease phenotype are not yet available [212].

In addition to the animal models mimicking the genetic changes observed in neuroblastomas, other transgenic murine models rely on the overexpression of virus-derived transforming genes [213]. They include mice overexpressing human adenovirus type 12E1A and E1B under the regulatory control of the mouse mammary tumor virus long terminal repeat [214], the early region of JC virus [215], and the hybrid of the metallothionein promoter–enhancer and the ret oncogene [216]. However, these models are less relevant to human disease and are not broadly used.

### 3.2. Syngeneic Murine Models

Syngeneic models of neuroblastoma utilize the mouse neuroblastoma cell line C1300 and its derivatives, such as Neuro-2A. These tumors possess immune and histological characteristics similar to human neuroblastomas. Because of that, they have been used successfully for testing chemotherapeutic strategies and immune-mediated approaches [188,217]. These models are inexpensive, easy to handle, and reproducible; however, they do not recapitulate the human cancer genetically, they demonstrate low cellular heterogeneity, and human immune cells are absent from them [16,218]. Consequently, they are considered as having a low relevance to human biology [117].

### 3.3. Xenograft Murine Models

Mouse xenografts are the most commonly used animal models of neuroblastoma [219,220,221]. This approach introduces human neuroblastoma cells into immunocompromised mice—typically athymic nude or severe combined immunodeficiency disease (SCID) mice. While these mice are excellent recipients of xenografts, the absence or partial absence of the immune system may restrict their use, e.g., for studies on immunotherapy [222]. The human cells can be introduced subcutaneously or orthotopically to the adrenal gland fat pad. While the first approach is easier to perform, it does not recapitulate the neuroblastoma tumor environment. Orthotopic injections, on the other hand, are technically challenging and require surgery to reach the adrenal gland.

Mouse xenografts are most commonly derived from established neuroblastoma cell lines, as they are widely available and easy to use. However, as described above, these cell lines are characterized by their own limitations, which may impact the results of the experiments in vivo. To overcome the issues related to the cell lines, patient-derived tissue fragments (or cells) are used for grafting [117,223] Using intact patient-derived tumors is preferable since it may bypass the alterations acquired by in vitro cultivation [224]. These tumors and cells are transplanted orthotopically or heterotopically into animals [58,225,226]. Importantly, to overcome a problem of scarcely available neuroblastoma patient-derived samples, cryopreserved implants may be used for generating xenografts [224]. The intact tumor patient-derived xenografts (PDXs) accurately recapitulate the human cancer microenvironment complexity, retain high-risk neuroblastoma features, such as high vascularization, as well as the presence of tumor-associated macrophages and cancer-associated fibroblast infiltration [227,228]. Overall, PDXs are considered to be of high relevance to human pathology [117] and can serve as in vivo models for the screening of potential therapeutic options, including immunotherapy [229] and virotherapy [230]. However, there are numerous limitations associated with these models, such as a high cost and the time-consuming labor involved in their maintenance, the limited availability of fresh tissues or PDXs, the modifications of the phenotype in cells derived from PDXs upon cell culture, and only around 50% of the engraftment success rate [109,225,231]. Nevertheless, the number of available neuroblastoma PDXs is growing. A large collection of well-characterized patient-derived tumors is now available through the Pediatric Preclinical Testing Consortium and Childhood Cancer Repository [232].

### 3.4. Humanized Mice and Other Future Murine Neuroblastoma Models

Importantly, xenograft models rely on immunocompromised animals that lack a functional immune system. To circumvent this issue, mice with a humanized immune system are being investigated as potential cancer models. These models require either an injection of human peripheral blood cells into mice, or the simultaneous injection of stromal tissue with tumor tissue [16,28,48,233]. However, thus far, no successful attempt with humanized mice models has been reported for neuroblastoma.

Mouse-human neural crest chimeras as neuroblastoma models were described in 2020 by Cohen et al. In this model, the human neural crest cells carrying genetic lesions observed in neuroblastomas were injected into a mouse embryo. This experimental approach resulted in the formation of human neuroblastomas in immunocompetent mice, accompanied by the potent infiltration by mouse immune cells [234]. Thus, in the future this approach may be considered a powerful model to test the immune aspects of neuroblastoma, potentially overcoming some issues related to traditional murine xenograft models developed in immunocompromised mice.

Other interesting attempts include personalized approaches, e.g., by generating a couple of different in vitro and in vivo models from the same patient-derived cells [235] that, taken together, may more closely represent the complexity of human disease. In another approach, treatment guided by patient personalized tumor grafts has been demonstrated [222]. However, the latter has not been tested for neuroblastomas.

**Table 2 cancers-15-03314-t002:** Preclinical in vivo models for neuroblastoma research, their major strengths, limitations, and human relevance.

Model	Strengths	Limitations	Human Relevance	References
Syngeneic mouse models	intact mouse immune system reproducible easy to handle low cost	do not accurately mimic biology and genetics of human disease no human immune cells present low cell heterogeneity	low	[16,117]
Transgenic mouse models	mimics biology, histology and genetics of human disease suitable to study neuroblastoma initiation	no metastases in typical locations expensive laborious	intermediate	[186,193,195,199,209]
Human xenograft murine models	recapitulates genetics of human disease easy to handle	require the use of immunocompromised animals do not accurately mimic progression of human disease no interaction with human stroma non-heterogenous, established cell lines used orthotopic models require surgery for cell injections	low	[219,220,221]
PDX models	accurately recapitulates human cancer heterogeneity and complexity retains high-risk neuroblastoma features	require the use of immunocompromised animals limited availability of tissues for transplantation costly laborious orthotopic models require surgery for cell implantation	high	[223,224,225,228]
Zebrafish models	low cost easy imaging	different temperature requirements for tumor cells and model cells requires specific equipment and tools no interaction with human stroma	low	[117,236]

### 3.5. Other Animal Models Proposed for Neuroblastoma Research

Except nude mice, nude rats have been utilized as potential models for neuroblastomas. However, these models did not exactly recapitulate the course of human malignancy [237]. Alternatively, some neural crest pathologies, including neuroblastoma, involving ALK activity and *LIN28B* gene expression were studied in the *Xenopus* model [238,239,240]. Zebrafish (*Danio rerio*) have been recently proposed as a promising platform to study genetics, pathogenesis, and progression processes in neuroblastoma [236,240,241,242,243]. Due to technical advantages over the mouse models, such as lower cost, shorter time to tumor development, and easy imaging, this model has become widely used in the neuroblastoma field. In addition, a recently established avian embryonic model involving grafting human neuroblastoma cells into chick embryos at the sympatho–adrenal crest level has been proven to be a valuable tool in recapitulating the interactions between normal developmental processes, the local microenvironment, and neuroblastoma progression [244,245]. The growth, differentiation, and potential drug sensitivity of neuroblastoma tumor can be also analyzed on the chick embryo chorioallantoic membrane [246,247,248]. However, even though the chick chorioallantoic membrane model is highly reproducible and easy to handle, it represents a low relevance to human pathophysiology [117].

### 3.6. Models of Neuroblastoma Metastasis

Modeling a metastatic disease has been a long-standing challenge in the field of neuroblastoma research. As described above, the tumors developing in transgenic mice typically do not form overt metastases. Similarly, the metastases from xenografts, including orthotopic tumors, are scarce and rarely seen in the niches relevant to human disease. Recapitulating osseous dissemination has proven to be particularly difficult. To overcome this problem, systemic injections to the tail vein are widely used [249,250]. Alternatively, intracardiac injections that facilitate bone colonization can be employed [251]. However, both models omit the initial stages of the metastatic process, which involve local invasion, intravasation, and escape from the primary tumor. On the other hand, direct injections of neuroblastoma cells into the bone cavity may be useful in investigations into the interactions between tumor cells and the bone environment, yet they do not recapitulate their dissemination [58]. This gap in our ability to model metastatic processes in neuroblastoma may be in the future filled by the use of PDXs. Recent years brought some reports of orthotopic PDXs metastasizing to the clinically relevant niches, such as bone, bone marrow, and liver [224]. However, as described above, the utility of these models is still hindered by the limited availability of neuroblastoma PDXs. Interestingly, a similar metastatic pattern has been shown upon orthotopic co-injection of established neuroblastoma cell lines with human mesenchymal stem cells [252]. Lastly, zebrafish models are useful to track neuroblastoma invasiveness; however, obtaining metastatic niche-specific information in this model is challenging.

## 4. Conclusions and Future Directions

Neuroblastoma research relies on preclinical models; however, no existing model is free of limitations. Therefore, it is of crucial importance to develop novel, advanced models that may serve as platforms for basic and translational research, especially for new treatment approaches. Conventional preclinical models in vitro, such as two-dimensional cell cultures, lack adequate representation of a tumor and do not recapitulate properly its biology, 3D architecture, topology, and many other features. Animal models, despite their importance in translational medicine and medical research fields, including neuroblastoma, also have serious disadvantages. They do not accurately recapitulate many aspects of human disease, such as its heterogeneity, the role of the immune system, and the complexity of the tumor microenvironment. The results of preclinical studies quite often do not predict the outcome in cancer patients; thus, the research on potential chemical agents does not proceed to clinical development. The gap between 2D cell cultures and animal models can be bridged by complex three-dimensional systems that closely recapitulate the course of human cancer [2,116,144,156,171]. Recent developments in three-dimensional, microfluidic, and multicellular systems used for neuroblastoma research [150] are promising, especially for testing new therapeutic approaches [118,160,253,254], including natural product research [177], immunotherapy [172,255,256], and new multimodal strategies for high-risk neuroblastoma [149]. A heterogenous disease such as neuroblastoma may largely benefit from comprehensive approaches involving a combination of various biological models supplemented by mathematical simulations [146,235,257,258]. Such strategies give promise for a better understanding of the malignancy, which may facilitate the development of effective therapies and improve the outcome for patients. However, it is important to note that currently available neuroblastoma models aim mainly at recapitulating the aggressive disease and do not reflect a full clinical spectrum of the disease. While this is justified by the need to test novel therapies for high-grade neuroblastoma, the basic biology of low-grade tumors and spontaneous regression, which is characteristic for these malignancies, remains understudied.

## Figures and Tables

**Figure 1 cancers-15-03314-f001:**
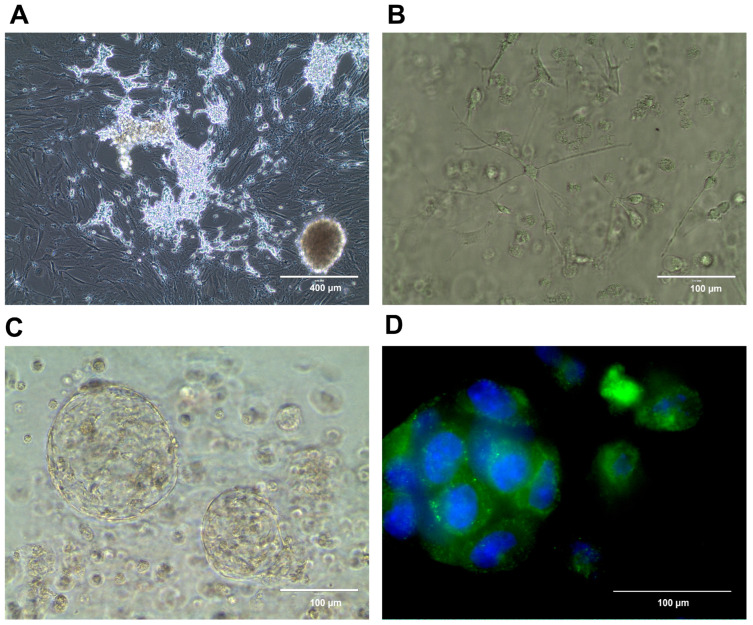
Conditionally reprogrammed murine neuroblastoma cells (CR-NB). Cells growing in a 2D cell culture, demonstrating an adrenergic cell population growing atop a mesenchymal cell population, as shown before [141] (**A**). The same CR-NB cell line in Matrigel 1 day (**B**) and 28 days of a 3D culture (**C**). The localization of neuropeptide Y in the 3D CR-NB structure (green); DNA (blue) stained using Hoechst 33258 (**D**).

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
