# Peer review of "Preclinical Models of Neuroblastoma—Current Status and Perspectives"

_cancers, 2023, doi:10.3390/cancers15133314_

Round 1

Reviewer 1 Report

This work constitutes and interesting and comprehensive summary of the available preclinical models to study neuroblastoma (NB). The authors do not get into too much detail about any of the models, but just summarize advantages and disadvantages from each of them. For example, I have missed that the authors comment on how good the models are when preserving the cellular heterogeneity of NB tumors. For example when discussing the utility of the different available cell lines, or when talking about mouse models such as TH-MYCN mouse. I would have liked to hear more about this aspect on available models. Otherwise, the review is sound and interesting, and only the following minor points can also be raised:

• At the beginning, some more recent reviews should be cited, in addition to Kamb, 2005. For example reviews like: Proietto et al., Front Oncol 2023; Fuochi and Galligioni, Methods Mol Biol. 2023; or Krüger and Kopp, Cancers (Basel) 2023.

• The initial title of the 3D section should be corrected to '3D cell cultures'.

• There is very scarce reference to studies on NB using 3D in vitro cultures. Same for models to study NB initiation. These two points should be a little more ellaborated, due to their potential future importance.

• Remove italics from the whole first paragraph of the xenograft Murine model section.

English is quite good. The authors just need to read it very carefully to detect the existing typos and correct them.

Reviewer 2 Report

With pleasure, I read the paper titled: “Preclinical models of neuroblastoma – current status and perspectives” by Krawczyk and Kitlińska. Overall, the subject matter is of clinical interest to a wide array of readers. The topic is intellectually relevant to the journal Cancers. Collectively, the manuscript reads well and has proper flow of ideas and data are summarized adequately in pertinent tables and figures. The main strength of the paper includes being an updated review on topic. I have the following comments/suggestions below.

1. Introduction. Please clearly highlight the significance of your work. It seems previous review reports have been published previously and therefore, it is critically central to emphasize the contemporary, meaningful impact of your present work.

2. It would be better if you authors can provide a table summary highlighting the major strengths, limitations, and human relevance of each in-vivo clinical model, including syngeneic mouse model, transgenic mouse model, PDX, zebra fish model, and human xenograft.

3. Neuroblastoma is a malignancy characterized by low rates of somatic genetic mutations. Instead, it is largely a disease of chromosomal alterations. Please provide a brief discussion on this matter and highlight how the preclinical models help or do not help to recapitulate these genetic chromosomal alterations.

4. Please provide a brief discussion on how these preclinical models help to study the role of immunotherapy in neuroblastoma.

5. Please provide a brief discussion on the benefits of preclinical models on studying the role of drug screen, as well as CRISPR/Cas9 screen.

6. For Table 1, it is highly recommended to add additional columns that describe the TP53 genetic status (wild-type or mutant), and predominant differentiation status based on gene expression or epigenetic profiling (epithelial or mesenchymal).

7. The authors may want to acknowledge the existence of the following high-risk genetic mouse model: TH-MYCN/ALK(F1174), TH-MYCN/Trp53(KI), and Dbh-iCre/CAG-C-MYC (the latter is cited in https://www.ncbi.nlm.nih.gov/pmc/articles/PMC8598007/).
